# Blood Plasma Exosomes Contain Circulating DNA in Their Crown

**DOI:** 10.3390/diagnostics12040854

**Published:** 2022-03-30

**Authors:** Oleg Tutanov, Tatiana Shtam, Alina Grigor’eva, Alexey Tupikin, Yuri Tsentalovich, Svetlana Tamkovich

**Affiliations:** 1V. Zelman Institute for Medicine and Psychology, Novosibirsk State University, 630090 Novosibirsk, Russia; ostutanov@gmail.com; 2Petersburg Nuclear Physics Institute Named by B.P. Konstantinov of National Research Center “Kurchatov Institute”, 188300 Gatchina, Russia; shtam_ta@pnpi.nrcki.ru; 3Institute of Chemical Biology and Fundamental Medicine, Siberian Branch of Russian Academy of Sciences, 630090 Novosibirsk, Russia; feabelit@mail.ru (A.G.); alenare@niboch.nsc.ru (A.T.); 4International Tomography Center, Siberian Branch of Russian Academy of Sciences, 630090 Novosibirsk, Russia; yura@tomo.nsc.ru

**Keywords:** circulating DNA, exosomes, DNA-binding proteins, histone-binding proteins, crown

## Abstract

It is known that circulating DNA (cirDNA) is protected from nuclease activity by proteins that form macromolecular complexes with DNA. In addition, it was previously shown that cirDNA can bind to the outer surface of exosomes. NTA analysis and real-time PCR show that exosomes from healthy females (HF) or breast cancer patients (BCP) plasma contain less than 1.4 × 10^−8^ pg of DNA. Thus, only a minor part of cirDNA is attached to the outer side of the exosome as part of the vesicle crown: the share of exosomal DNA does not exceed 0.025% HF plasma DNA and 0.004% BCP plasma DNA. Treatment of plasma exosomes with DNase I with subsequent dot immunoassay reveals that H2a, H2b, and H3 histones are not part of the exosomal membrane, but are part of the cirDNA–protein macromolecular complex associated with the surface of the exosome either through interaction with DNA-binding proteins or with histone-binding proteins. Using bioinformatics approaches after identification by MALDI-TOF mass spectrometry, 16 exosomal DNA-binding proteins were identified. It was shown that four proteins—AIFM1, IGHM, CHD5, and KCNIP3—are candidates for DNA binding on the outer membrane of exosomes; the crown of exosomes may include five DNA-binding proteins: H2a, H2b, H3, IGHM, and ALB. Of note, AIFM1, IGHM, and CHD5 proteins are found only in HF plasma exosomes; KCNIP3 protein is identified only in BCP plasma exosomes; and H2a, H2b, H3, and ALB are revealed in all samples of plasma exosomes. Two histone-binding proteins, CHD5 and KDM6B, have been found in exosomes from HF plasma. The data obtained indicate that cirDNA preferentially binds to the outer membrane of exosomes by association with DNA-binding proteins.

## 1. Introduction

Circulating nucleic acids were discovered in 1948, but interest in them has increased significantly in the past two decades. All characteristic signs of genomic DNA from cancer cells have been detected in blood cirDNA; these include point mutations [1], altered microsatellite composition [2], a characteristic methylation profile of tumor suppressor genes [3,4] and repeats [5], and deletions and chromosomal rearrangements [6]. With the development of methods of molecular biology and biochemistry, significant progress has been made in the use of circulating DNA (cirDNA) in practical medicine [7,8,9,10]. In particular, in 2016, the FDA approved the Cobas EGFR Mutation Test v2 (Roche Molecular Systems, Inc., Pleasanton, CA, USA) for the diagnosis of somatic activating mutations in the *EGFR* gene in the plasma of patients with non-small cell lung cancer. Other promising markers of tumor transformation are epigenetic changes in the composition of cirDNA, which primarily include the methylation of cytosines at the fifth position in CpG dinucleotides. For example, the analysis of the methylation of the genes Septin 9 and SHOX2 in blood plasma cirDNA allows for detecting intestinal and lung cancer with a sensitivity and a specificity greater than 90% [11,12]. However, at present, there is still no unequivocal opinion regarding the sources of cirDNA, the mechanisms that ensure their circulation and elimination from the bloodstream, and, most importantly, the biological role of cirDNA [13,14,15]. It is known that cirDNA is resistant to the action of extracellular nucleases due to the formation of macromolecular complexes with proteins (including histones) [16] and other biomolecules, as well as the binding of these complexes to the surface of blood cells [17,18]. Notably, several pathological conditions, including cancer and autoimmune diseases, which are characterized by increased cirDNA concentrations [19,20], are also characterized by elevated exosome levels in plasma or serum [21]. Despite their small size, exosomes have a large surface area, and their membrane largely reflects that of the parent cells; thus, by analogy with cell-surface-bound DNA, they are able to transport DNA at least on their surface. Taking into account the content of exosomes in the blood (about 10^7^–10^8^ vesicles/mL), we hypothesized a significant role of this type of vesicles in cirDNA blood transportation, DNA protection against plasma nucleases, and targeted delivery of cirDNA to recipient cells. Notably, it was found earlier that EV surface-associated DNA is involved in horizontal gene transfer [22,23] and induction of autoimmunity [24]. In the future, capturing the biological implications of secretion of exosome in different pathological conditions could be useful for further describing innovative biomarker nanotechnological platforms to be applied to diseases in which exosomes play a role, greatly impacting the new era of liquid biopsy [25].

Here, we confirm the presence of surface-associated DNA on exosomes, determine the share of exosomal DNA (exoDNA) in the plasma cirDNA of healthy females (HFs) and breast cancer patients (BCPs), identify proteins involved in the association of exoDNA with the outlier exosomal membrane, and propose mechanisms for this association.

## 2. Materials and Methods

### 2.1. Blood Treatment and Exosome Isolation

Blood samples from HFs (*n* = 7, age 32–56 years, median age 41) were obtained from Novosibirsk Central Clinical Hospital; blood samples from previously untreated luminal BCPs (*n* = 7, age 33–61 years, median age 57) were obtained from Novosibirsk Regional Oncology Dispensary (Table 1).

Venous blood (9 mL) was collected in K_3_EDTA spray-coated vacutainers (Improvacuter, China, cat. no. 694091210), immediately mixed using a rotary mixer, placed at +4 °C, and fractionated into plasma and blood cells within an hour after blood sampling.

Blood was centrifuged at 290× *g* for 20 min. Then, blood plasma was transferred into a new tube and centrifuged a second time at 1200× *g* for 20 min. Plasma was aliquoted and kept on ice or +4 °C until exosome isolation to avoid any freezing of the samples. Samples that were used for estimation of cirDNA in plasma were stored at −80 °C until measurement.

For isolation of exosomes, plasma was centrifuged at 17,000× *g* for 15 min to pellet cell debris. Supernatant of plasma was diluted 1:5 by PBS, and all fractions were passed through a 100 nm pore size filter (Minisart high flow, 16553-K, Sartorius, Goettingen, Germany).

The filtrate was then centrifuged for 90 min at 100,000× *g* at 4 °C. The pellets were suspended in 12 mL of PBS and again centrifuged for 90 min at 100,000× *g* at 4 °C. The washing stage was repeated two times. Then, supernatant was removed, and pellets were resuspended in 250 μL of PBS. To study the location of exosomal DNA (exoDNA), half of the exosome samples from different individuals were mixed to generate one sample from HFs and one sample from BCPs. Individual and mixed samples were frozen in liquid nitrogen and stored in aliquots at −80 °C. The aliquots were thawed once before use.

### 2.2. Exosome Characterization

Morphology of the isolated extracellular vesicles (EVs) was assessed by transmission electron microscopy (TEM) as described previously [26]. Specific markers for exosomes, CD9 or CD63 were detected as described earlier [26] using 10 μL of vesicles mixed with 10 μL of 0.5% bovine serum albumin in PBS and 3 μL (100 μg/mL) of corresponding monoclonal antibodies (Abcam, Cambridge, UK). All grids were examined in JEM 1400 (Jeol, Tokyo, Japan) TEM supplied with digital camera Veleta (EM SIS, Muenster, Germany). The measurements were made directly on the camera screen using iTEM (EM SIS, Muenster, Germany) software, version 5.2.

The size and concentration of the isolated EVs were determined by NTA using the NanoSight^®^ LM10 (Malvern Instruments, Malvern, Worcestershire, UK) analyzer, equipped with a blue laser (45 mW at 488 nm) and a C11440-5B camera (Hamamatsu Photonics K.K., Shizuoka, Japan), at several dilutions according to the manufacturer’s instructions. Each sample was measured in triplicate, with a camera setting of 15, an acquisition time of 60 s, and a detection threshold setting of 5. At least 200 completed tracks were analyzed per video. NTA analytical software version 2.3 (Malvern Instruments, Malvern, Worcestershire, UK) was used for data analysis and capture.

### 2.3. DNA Isolation and Quantification

CirDNA was isolated from 1 mL of blood plasma and from 100 μL of plasma exosomes using the “DNA Isolation Kit” (BioSilica Ltd., Novosibirsk, Russia) according to the manufacturer protocols and concentrated by precipitation in acetone as triethylammonium salts to 15 μL [27]. Concentration of isolated DNA was measured by the TaqMan real-time PCR of human L1 fragments (Q-PCR) [27]. Genomic DNA from human leukocytes served as a standard for obtaining the calibration curves. The Q-PCR was performed with an ICycler iQ5 (Bio-Rad, Hercules, CA, USA). The DNA concentration was estimated according to the initial volume of each blood sample.

The sizes of cirDNA and exoDNA were evaluated using an Agilent High Sensitivity DNA Kit and Bioanalyzer (Agilent Technologies, Waldbronn, Germany).

### 2.4. Examination of exoDNA Topology

To study the location of DNA in the exosomes and to analyze the susceptibility to DNase I digestion, 460 μL of non-frizzed plasma exosomes was either pre-treated for 30 min at 37 °C with 8 μL of DNase I (1 e.u./mL, without RNase activity, Fermentas, Vilnius, Lithuania), with 52 μL of 10× DNase buffer, or mock treated with PBS in DNase buffer. After incubation for 30 min at 37 °C, DNase was inhibited by EDTA and heating (65 °C, 10 min). After DNase I treatment, the samples were washed by 12 mL of PBS and ultracentrifuged (100,000× *g*, 4 °C, 90 min); supernatants and exosomes were collected for DNA isolation and subsequent evaluation of sizes and concentrations.

### 2.5. Dot Immunoassay

The 0.5 μL of isolated exosomes or treated by DNase I exosomes were absorbed on a D-3354 nitrocellulose membrane (Schleicher&Schuel, Dassel, Germany) and dried for 20 min at room temperature. Nonspecific sorption was blocked with the buffer (10 mM Tris-HCl, pH 7.5, 0.15 M NaCl, 10% of bovine serum) on a shaker at 700 rpm for 20 min. A solution of rabbit polyclonal antibodies against H1, H2A, H2B, and H3 histones (2.5 μg/mL, 100 μL of each) or mouse monoclonal antibodies against CD9 (Abcam) in the working buffer (10 mM Tris-HCl, pH 7.5, and 0.15 M NaCl, 10% of bovine serum, 0.05% Tween-20) was added to the nitrocellulose dried at room temperature for 20 min and the mixture was incubated on a shaker for 4 h at 700 rpm. Antigen-unbound antibodies were washed out 5 × 2 min with the working buffer, and goat anti-rabbit antibodies or rabbit anti-mouse antibodies conjugated with horseradish peroxidase (Abcam) diluted (1:5000) in the same buffer were added. The incubation with the conjugate was performed for 4 h under shaking (700 rpm). The wells with the samples were washed, and antigen–antibody complexes were visualized [28].

### 2.6. Identification of Exosomal Proteins by MALDI-TOF Mass-Spectrometry

To determine the primary structure of proteins in the plasma exosomes of HFs and BCPs, proteins were separated using 10–20% SDS-disk electrophoresis and stained with Coomassie R-250 (Sigma, St. Louis, MO, USA). Fragments of polyacrylamide gel containing the studied proteins were washed from Coomassie R-250 and SDS and subjected to trypsinolysis as described previously [28]. Peptide fragments of proteins were extracted from the gel, concentrated, and desalted on C18 ZipTips microcolumns (Milipore, Burlington, MA, USA). The mixture of peptides was eluted from the microcolumn onto the instrument plate target with a saturated matrix solution.

The acquisition and registration of mass spectra was carried out on a time-of-flight tandem mass spectrometer MALDI-TOF autoflex speed series LIFT (Bruker Daltonics, Bremen, Germany) at the Mass Spectrometry Research Center of the Siberian Branch of the Russian Academy of Sciences. Protein identification was performed by searching for appropriate candidates in the annotated NCBI and SwissProt databases using the Mascot program (Matrix Science Ltd., London, UK, www.matrixscience.com/search_form_select.html accessed on 25 February 2022) as described in [28].

### 2.7. Data Analysis

Statistical analysis was performed using the Statistica 6.0 software. All the data were expressed as medians with interquartile ranges; *p*-values < 0.05 were considered to be statistically significant.

Analysis of the universality of exosome proteins and the comparison of the identified proteins with the proteins annotated in the Vesiclepedia database (www.microvesicles.org accessed on 25 February 2022) were performed using the FunRich software (version 3.1.4 software package) (http://www.funrich.org accessed on 25 February 2022). Analysis of the presence of DNA-binding domains in exosomal proteins was carried out using the Interpro web platform (https://www.ebi.ac.uk/interpro/ accessed on 25 February 2022), PROSITE, Pfam databases (http://pfam.xfam.org/ accessed on 25 February 2022), SMART (http://smart.embl-heidelberg.de/ accessed on 25 February 2022), and CDD (https://www.ncbi.nlm.nih.gov/Structure/cdd/cdd.shtml accessed on 25 February 2022) [29].

## 3. Results

### 3.1. Characterization of Exosomes

To characterize EVs isolated from blood plasma, TEM combined with immunogold detection of CD63 and CD9 onto EV surface and NTA were used. TEM revealed the presence of clearly structured cup-shaped vesicles (40–100 nm) of low electron density with a preserved membrane in all samples: plasma exosomes from HF blood and plasma exosomes from BCP blood. Vesicles with damaged membranes did not exceed 15%, and the proportion of microvesicles (with the size smaller than 30 nm) was no more than 20% (Figure 1a).

Various exosome types circulate in the blood of healthy donors and patients with different cancers. CD63 and CD9 are essential structural components of exosome membranes that mediate adhesion of exosomes to the surface of the recipient cell [30]. Immunogold labeling of plasma exosomes in blood of HFs and BCPs revealed CD9 and CD63 at the surface of isolated vesicles (Figure 1b).

NTA revealed that EVs from HF blood and BCP blood had similar sizes: HF plasma EVs had a mean size of 104 nm, with a mode of 74 nm and an SD of 57 nm (Figure 2a); BCP plasma EVs had a mean size of 109 nm, with a mode of 89 nm and an SD of 65 nm (Figure 2b). However, an increased concentration of exosomes in the blood plasma of cancer patients was revealed in comparison with HFs (median concentration 18 × 10^7^ and 7 × 10^7^ EVs/mL of blood, respectively).

Collectively, these data demonstrate that vesicles isolated from plasma demonstrate all characteristics of exosomes.

### 3.2. Circulating DNA in Blood Plasma

The cirDNA concentration in the plasma of healthy women and BCPs was estimated by Q-PCR. A significant increase in the plasma DNA concentration was found for BCPs compared with HFs (median 46 versus 4 ng/mL, *p* = 0.0026, Mann–Whitney U-test) (Figure 3a). The revealed difference between the tumor patients and the controls coincides with published data [17,31].

Whereas the cirDNA concentration in samples was quite low, individual samples of plasma DNA were pooled. It was found that the HF plasma samples mainly contained DNA fragments ~180 bp and fragments 320, 540, and more than 7000 bp to a much smaller extent. Mainly high-molecular-weight DNA (more than 2000 bp) was found in plasma DNA from BCPs (Figure 3b). Obtained results coincide with earlier published data [32,33].

### 3.3. DNA Cargo of Blood Exosomes

The median concentration of exoDNA in plasma of BCPs was found to be about 2 pg/mL of blood (DNA was detected in 71% (5/7) samples with range 0–7 pg/mL) and in plasma of HFs, it was found to be about 1 pg/mL of blood (DNA was detected in 57% (4/7) of samples with a range of 0–29 pg/mL) (Figure 4), similar to published data [34]. Thus, one exosome contains 1.4 × 10^−8^ pg of DNA in HF plasma, and in BCP plasma, 1.1 × 10^−8^ pg of DNA. Only a minor part of cirDNA is attached to the outer side of the exosome as part of the vesicle crown—the share of exoDNA does not exceed 0.025% HF plasma DNA and 0.004% BCP plasma DNA.

Taking into account that the internal volume of one exosome is approximately 4 × 10^−24^ to 1.5 × 10^−21^ m^3^ and the volume of one average 50 kDa protein or 100 nt RNA molecule is approximately 6 × 10^−26^ m^3^ [34], theoretical calculation allows us to assume that each exosome can accommodate approximately 50,000 molecules of proteins or small RNA. Since DNA of high molecular weight (2–7 kb) were detected predominantly in the blood plasma of BCPs (Figure 3b), the calculation appears to show that the average exosome can contain no more than 700 molecules of DNA. However, since exosomes have been shown to predominantly contain small RNAs and proteins, the actual number of DNA molecules in an exosome is even smaller. Indeed, significantly lower quantities of DNA were obtained from the experiments. In particular, if the molecular weight of 5 kbp DNA is 3.45 × 10^6^ and the quantity of exoDNA is 1.1 × 10^−8^ pg, then each exosome should contain 1.9 × 10^−3^ DNA molecules.

Previously, we demonstrated that the blood of healthy donors, as well as that of patients with lung cancer [35], stomach and colon cancers [36], prostate cancer [37], and breast cancer [38] constantly contains cirDNAs that occur not only in blood plasma, but also in complexes bound to the surface of blood cells. A portion of cell surface-bound cirDNA (csbDNA) dissociates after the treatment of cells with PBS-EDTA buffer and is apparently bound to phospholipids and other anions of the cell membrane through bridges of divalent metal ions or low-affinity interactions and is eluted with nine buffer volumes (compared to the plasma volume); another csbDNA portion is removed from the cell surface by treating cells with a 0.125% trypsin solution and, apparently, is a part of the complexes with surface proteins of the blood cells [17]. Since the surface of an exosome carries DNA-binding proteins similar to the parent cell, we hypothesized the possibility of circulating DNA to binding with the external surface of exosomes. Treatment of the exosomes with DNase I and mock treatment of exosomes with PBS in DNase buffer revealed that DNA from HFs’ exosomes and DNA from BCPs’ exosomes were fully accessible for cleavage with the enzyme. Obtained results indicate that DNA is located on the outer surface of exosomes and that it is insufficiently protected by proteins from nuclease activity.

The presence of H2a, H2b, and H3 histones and CD9 in untreated exosomes from plasma of HFs and BCPs was confirmed using commercial antibodies; only CD9 was revealed in exosomes treated by DNase I (Figure 5).

According to the obtained data, histone H1 was absent from the surface of exosomes both untreated and treated by nuclease. Moreover, the presence of CD9 and the disappearance of H2a, H2b, and H3 histones after treatment with DNase I indicate that these histones are not part of the exosomal membrane, but are part of the cirDNA–protein macromolecular complex associated with the surface of the exosome either through interaction with DNA-binding proteins or with histone-binding proteins.

### 3.4. Identification of DNA-Binding and Histone-Binding Proteins in Plasma Exosomes

Using MALDI-TOF mass spectrometry, 85 and 80 proteins were identified with high reliability (*p* < 0.05) in the blood plasma exosomes of HFs and BCPs, respectively (Appendix A). Of these, only 20% were common for both groups (Appendix A). Histones H2a, H2b, and H3 were included in the analysis but were not confirmed by MALDI-TOF due to their high Lys contents and consequent excessive trypsinolysis.

Of the proteins we identified in plasma exosomes, 74% have already been detected in the composition of EVs by mass spectrometry, and they are annotated in the Vesiclepedia database (www.microvesicles.org accessed on 25 February 2022). Thus, despite active research in the field of EVs in general and exosomes in particular, we have identified 35 vesicular proteins for the first time (Appendix A).

At the next stage, to identify DNA-binding proteins, proteomes of exosomes from the blood plasma of HFs and BCPs were analyzed using QuickGO (https://www.ebi.ac.uk/QuickGO/annotations accessed on 25 February 2022) (“DNA binding”, GO:0003677; “double-stranded DNA binding”, GO:0003690; “RNA polymerase II cis-regulatory region sequence-specific DNA binding”, GO:0000978; “chromatin DNA binding”, GO:0031490; “sequence-specific DNA binding”, GO:0043565; “double-stranded DNA binding”, GO:0003690; “satellite DNA binding”, GO:0003696; “RNA polymerase II transcription regulatory region sequence-specific DNA binding”, GO:0000977). It was shown that 14 (17%) and eight (10%) proteins in plasma exosomes of HFs and BCPs are associated with DNA binding, respectively, of which six (21%) proteins were universal (Figure 6).

Several DNA-binding domains have been identified in DNA-binding exosomal proteins (Table 2). 

According to a recently published study on the binding of transcription factors to nucleosomes based on the ratio of Kd for non-specific DNA binding to Kd for non-specific binding of nucleosomes, as well as the three-dimensional structure of their DNA-binding domains, it is possible to conditionally divide DNA-binding proteins into “strongly” and “weakly” binding nucleosomes [39]. The DNA-binding proteins that “strongly” bind nucleosomes were characterized by the presence of the following types of domains: bHLH, HTH, homeodomains, and zinc fingers. Thus, among the 13 DNA-binding exosomal proteins, excluding histones, seven (54%) have a high affinity for nucleosome binding (one universal protein; one and five proteins in BCP plasma exosomes and HF plasma exosomes, respectively) and can be the main structural elements of nucleoprotein complexes on the outer membrane of exosomes.

Analysis of cellular localization of exosomal proteins via QuickGO has shown that four proteins—AIFM1 (“membrane”, GO:0016020), IGHM (“membrane”, GO:0016020; “integral component of membrane”, GO:0016021; “plasma membrane”, GO:0005886), CHD5 (“membrane”, GO:0016020), and KCNIP3 (“membrane”, GO:0016020; “plasma membrane”, GO:0005886; “voltage-gated potassium channel complex”, GO:0008076)—are candidates for DNA binding on the outer membrane of exosomes; the crown of exosomes may include five DNA-binding proteins: H2a, H2b, H3 (in according our dot-analysis results), IGHM, and ALB. Of note, AIFM1, IGHM, and CHD5 proteins are found only in HF plasma exosomes; KCNIP3 protein is identified only in BCP plasma exosomes; and H2a, H2b, H3, and ALB are revealed in all samples of exosomes (Table 2).

In addition to binding cirDNA to DNA-binding proteins of the exosome membrane, cirDNA can be attached to exosomes by binding histones in nucleoprotein complexes to histone-binding proteins of the exosomal crown. However, only two histone-binding proteins, CHD5 and KDM6B, have been found in plasma HF exosomes. The data obtained indicate that cirDNA preferentially binds to the outer membrane of exosomes by association with DNA-binding proteins.

## 4. Discussion

It is known that the cirDNA concentration in the blood increases in various diseases, including cancer [40]. At the same time, the cirDNA concentration in the cancer patients’ blood correlates with the activity of extracellular DNases [37], as well as extracellular proteases that destroy proteins of circulating DNA–protein complexes [19,41]. At present, in addition to histones [16], more than 150 proteins involved in the formation of nucleoprotein complexes in plasma have been identified [42,43,44].

Binding of nucleoprotein complexes to the surface of EVs may serve as an additional mechanism that ensures long-term circulation of DNA in the blood. Indeed, a number of studies have shown the presence of DNA on the outer surface of the exosomal membrane [45,46]. However, it is still unclear what proportion of cirDNA might circulate in association with exosomes.

In this work, the presence of histones in the crown of exosomes was confirmed and the amount of cirDNA associated with exosomes was estimated. Since the share of exoDNA in the crown of exosomes does not exceed 0.025% HF plasma DNA and 0.004% BCP plasma DNA, exoDNA has an extremely low diagnostic value, although this is speculated in many reviews. At the same time, cirDNA in nucleoprotein complexes as part of the protein crown of exosomes can be successfully targeted, transported to recipient cells, and, along with miRNAs and proteins, might have important biological significance.

Several variants of DNA binding to the cell surface have been described. It has been found that almost all cells are able to bind DNA (or oligonucleotides) and that DNA can bind to membranes both with the help of membrane receptors and without involvement of membrane proteins. These receptors can be polyanion receptors [47], nonspecific receptors (scavenger receptors) [48], or specialized receptors that predominantly bind nucleic acids [49]. Numerous cell surface proteins that are able to bind DNA have been discovered and partially identified [50]. Since the membrane of exosomes largely resembles the membrane of the parent cell, similar mechanisms of cirDNA binding to the exosome can be assumed. Indeed, four DNA-binding proteins of membrane localization (AIFM1, IGHM, CHD5, KCNIP3) have been identified in exosomes in the current work. In addition, five DNA-binding proteins of extracellular localization (IGHM, ALB, H2a, H2b, H3) were identified in exosomes. These data indirectly indicate that ALB, H2a, H2b, and H3 proteins bind cirDNA, and, in turn, are associated with other proteins on the outlier surface of exosomes.

One striking example of these proteins is the histone-binding protein CHD5 identified in the current work in the exosomal membrane. The disappearance of H2a, H2b, and H3 histones after treatment with DNase I indicates that these histones are not a part of the exosomal membrane, but are a part of the cirDNA–protein macromolecular complex associated with the surface of exosomes either through interaction with DNA-binding proteins or with histone-binding protein CHD5. The absence of H1 histone at the exosomal crown can be explained by its substitution with amyloid P for increased oligonucleosome solubility in plasma [43].

Moreover, although there are no DNA-binding Gene Ontology terms yet assigned to some proteins, the literature evidence suggests that there are other DNA-binding proteins in exosomal proteomes. Notably, fibrinogen has already been shown earlier to bind cirDNA [42]. Considering that it has also been shown to be localized in the crown of vesicles, forming complexes with integrins, the formation of macromolecular complexes of DNA–fibrinogen–integrin–exosome seems quite possible [45].

In addition to DNA-binding extracellular proteins and membrane proteins, 13 and eight DNA-binding proteins of nuclear and cytoplasmic origin were identified in plasma exosomes of HFs and BCPs, respectively. It was shown that these exosomal DNA-binding proteins have many molecular functions and are involved in important biological processes [51,52,53,54]. Thus, exosomes carry biologically active DNA-binding proteins mainly in the form of internal contents from donor cells to recipient cells, causing changes in the behavior of recipient cells. The biological significance of such small amounts of cirDNA on the surface of exosomes is still a mystery.

## Figures and Tables

**Figure 1 diagnostics-12-00854-f001:**
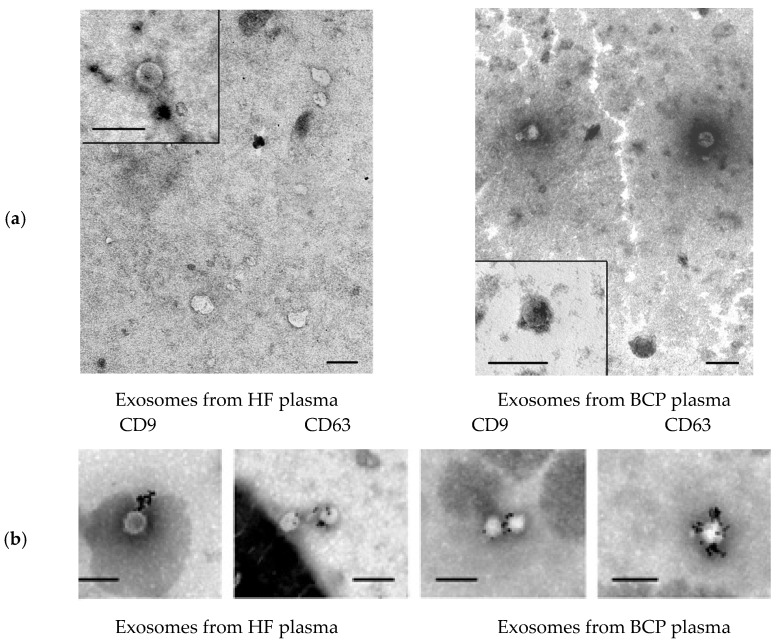
Electron microscopy. (**a**) Total view of exosome preparation obtained from blood plasma of HFs and blood plasma of BCPs. Inserts show exosomes. Scale bars correspond to 100 nm, negative staining by phosphotungstate acid. (**b**) Immunogold labeling of plasma exosomes in blood of HFs and BCPs based on electron microscopy data. Exosomes were incubated with monoclonal antibodies to CD63 or CD9 with subsequent detection by conjugate of protein A and gold nanoparticles. Scale bars correspond to 100 nm. Negative staining is shown by phosphotungstic acid.

**Figure 2 diagnostics-12-00854-f002:**
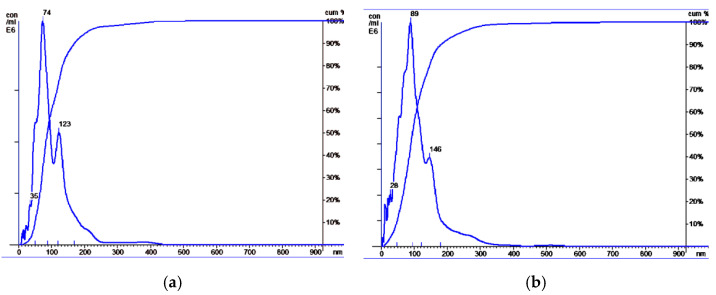
Size distribution of plasma exosomes isolated from the blood of HFs and BCPs. Data of NTA analysis (Malvern, NS-300). (**a**) Exosomes from blood plasma of HFs; (**b**) exosomes from blood plasma of BCPs.

**Figure 3 diagnostics-12-00854-f003:**
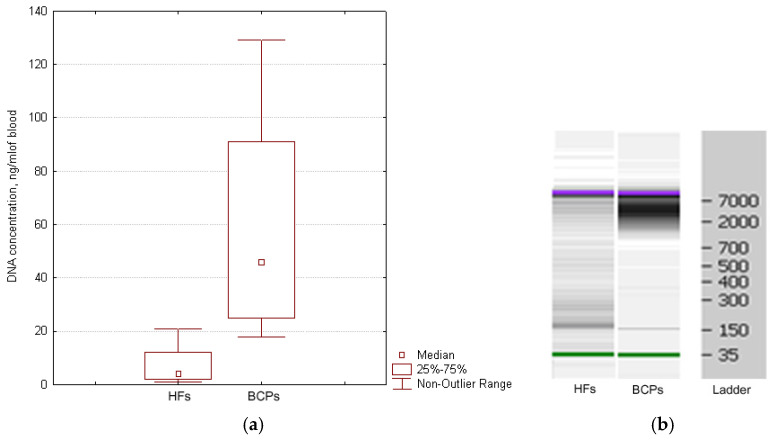
Characterization of cirDNA from the plasma of HFs and BCPs. (**a**) DNA quantification from the plasma of HFs and BCPs. Tukey box plots of cirDNA. Median cirDNA concentration with 25–75% and non-outlier range bars are indicated. (**b**) Size distribution of cirDNA extracted blood plasma of HFs and BCPs. The data from Agilent 2100 Bioanalyzer^TM^ with 35 nt and 10,380 nt DNA fragments as an internal standards are shown.

**Figure 4 diagnostics-12-00854-f004:**
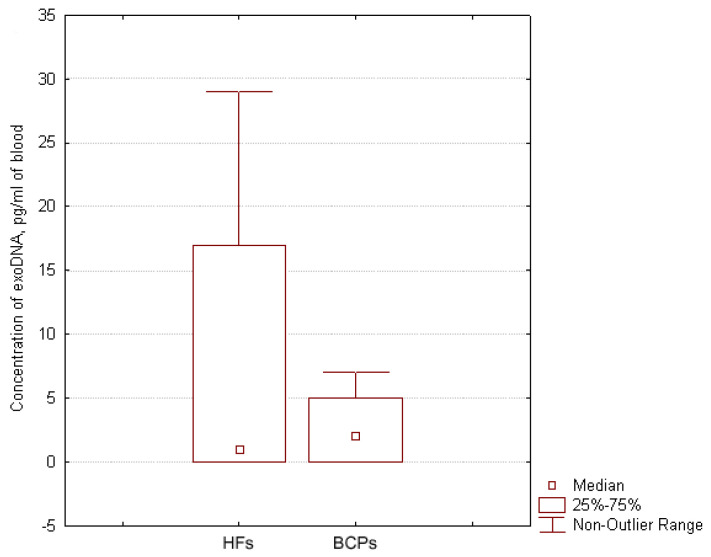
Exosomal DNA concentration in the blood of HFs and BCPs. Tukey box plots of exoDNA. Median exoDNA concentrations with 25–75% and non-outlier range bars are indicated.

**Figure 5 diagnostics-12-00854-f005:**
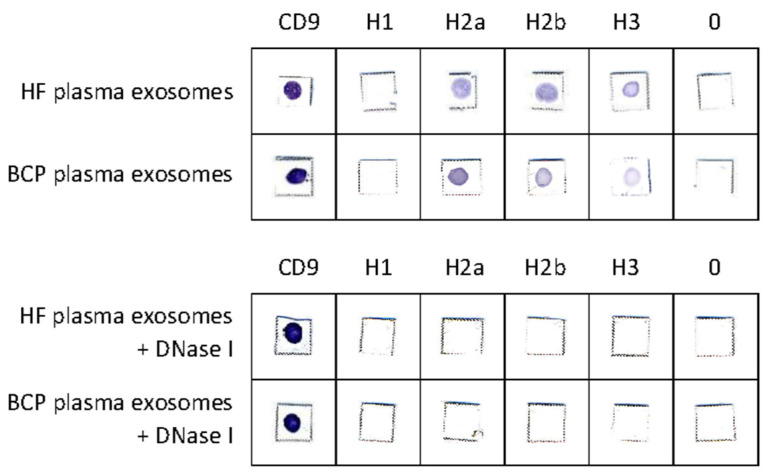
Dot immunoassay. The exosomes from plasma of HFs and BCPs were absorbed on a nitrocellulose membrane and incubated with antihistone or anti-CD9 antibodies and then incubated with a conjugate of goat anti-rabbit antibodies or rabbit anti-mouse antibodies conjugated with horseradish peroxidase. The antigen–antibody complexes were visualized with 4-chloro-1-naphtol.

**Figure 6 diagnostics-12-00854-f006:**
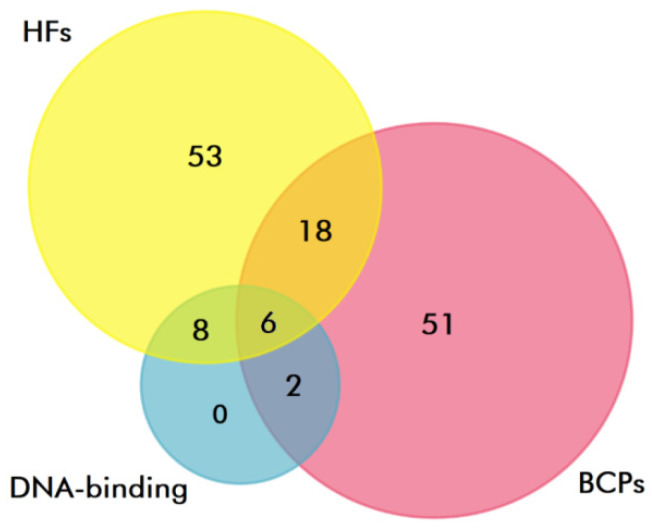
Venn–Euler diagram of DNA-binding proteins in plasma exosomes from HF and BCP blood, composed using QuickGO and FunRich software.

**Table 1 diagnostics-12-00854-t001:** General clinical characteristics of BCPs.

		No (%)
Tumor Stage	T1	4 (57%)
	T2	3 (43%)
Nodal Status	N0	7 (100%)
	M0	7 (100%)
ER and Pr Receptor Status	Positive	7 (100%)
HER2 Status	Positive	7 (100%)
Infiltrative Ductal Carcinoma		7 (100%)
Total Patients		7 (100%)

**Table 2 diagnostics-12-00854-t002:** DNA-binding exosomal proteins.

Uniprot ID	Gene Name	Protein Name	DNA-Binding Domain	Source of Exosomes	Score	Peptides Matched
O95831	AIFM1	Apoptosis-inducing factor 1, mitochondrial	FAD/NAD(P)-binding domain	*HFs*	59	15
O14862	AIM2	Interferon-inducible protein AIM2	N/A	*HFs*	57	5
Q8TDI0	CHD5	Chromodomain-helicase-DNA-binding protein 5	CHD subfamily II, SANT-like domain; CHD, C-terminal 2; Zinc finger, PHD-type; Zinc finger, PHD-finger	*HFs*	47	17
O96004	HAND1	Heart- and neural crest derivatives-expressed protein 1	basic helix-loop-helix (bHLH) domain	*HFs*	61	6
P01871	IGHM	Ig mu chain C region	Immunoglobulin-like domain	*HFs*	75	11
P40938	RFC3	Replication factor C subunit 3	N/A	*HFs*	64	25
Q9Y2P0	ZNF835	Zinc finger protein 835	Zinc finger C2H2-type	*HFs*	110	10
O43309	ZSCAN12	Zinc finger and SCAN domain-containing protein 12	Zinc finger C2H2-type	*HFs*	57	7
P02768	ALB	Serum albumin	N/A	*HFs, BCPs*	62	23
O75531	BANF1	Barrier-to-autointegration factor	2 non-specific dsDNA-binding sites	*HFs, BCPs*	60	5
Q15776	ZKSCAN8	Zinc finger protein with KRAB and SCAN domains 8	Zinc finger C2H2-type	*HF, BCPs*	55	12
Q9Y2W7	KCNIP3	Calsenilin	EF-hand domain	*BCPs*	58	8
Q14966	ZNF638	Zinc finger protein 638	Matrin/U1-C, C2H2-type zinc finger; RNA recognition motif domain	*BCPs*	65	46

## Data Availability

The data presented in this study are available on request from the corresponding author.

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
