# Peer review of "Blood Plasma Exosomes Contain Circulating DNA in Their Crown"

_diagnostics, 2022, doi:10.3390/diagnostics12040854_

Round 1

Reviewer 1 Report

The article of Tutanov and colleagues is mainly devoted to the discovery of the mechanism of interaction between exosomes and circulating DNA (cirDNA). The study was well conducted and the results are clearly demonstrated.
Also, the methodology section is well described and doesn't need particular revision.  
A point that could be improved is related to the introductive part of the paper including the impact of exosomes in disease progression (see as ex. PMID: 32932746) and the need for new technologies able to associate a specific marker with an exosome subtype and this exosome subtype to a particular function and/or group of functions (PMID: 35141731 is just an example). The authors should explain better the intent of their work. 

Others:

The conducted research describes an interesting approach to discovering the complex association between exosomes and circulating DNA (cirDNA) that for sure have an impact in the field.

The well-performed MALDI-TOF analysis  is something that adds really informative data to the literature.

The conclusions are completely consistent with the results.
As highlighted in the already updated comments, the references could be improved with some more recent publications.

Figures look substantially nice but the related captions should be more informative in order methods and results.

Good luck!

Author Response

Thank you very much for your kind letter, as well as the enclosed suggestions and comments, regarding our manuscript entitled “Blood Plasma Exosomes Contain Circulating DNA in Their Crown”. We thank you for providing critical reviews and helpful suggestions regarding our submitted manuscript. Accordingly, we have carefully revised the paper, highlighted changes within the manuscript, with changes tracked. Please see herein a detailed response to the comments provided, point-by-point, reflecting changes made within the manuscript.

A point that could be improved is related to the introductive part of the paper including the impact of exosomes in disease progression (see as ex. PMID: 32932746) and the need for new technologies able to associate a specific marker with an exosome subtype and this exosome subtype to a particular function and/or group of functions (PMID: 35141731 is just an example).

Response: We agree with the opinion of the reviewer and expanded the Introduction by adding perspectives on the use of exosome contents for the development of liquid biopsy methods with citation of manuscript:

Maisano, D.; Mimmi, S.; Dattilo, V.; Marino, F.; Gentile, M.; Vecchio, E.; Fiume, G.; Nisticò, N.; Aloisio, A.; de Santo, M.P.; Desiderio, G.; Musolino, V.; Nucera, S.; Sbrana, F.; Andò, S.; Ferrero, S.; Morandi, A.; Bertoni, F.; Quinto, I.; Iaccino, E. A Novel Phage Display Based Platform for Exosome Diversity Characterization. Nanoscale. 2022, 14(8), 2998-3003.

The authors should explain better the intent of their work. 

Response: We have tried to define the goal of our research more specifically by correcting the previous text to the following:

Here we confirm the presence of surface-associated DNA on exosomes, determine the share of exosomal DNA (exoDNA) in the plasma cirDNA of healthy females (HFs) and breast cancer patients (BCPs), identify proteins involved in the association of exoDNA with the outlier exosomal membrane, and propose mechanisms for such association.

The conducted research describes an interesting approach to discovering the complex association between exosomes and circulating DNA (cirDNA) that for sure have an impact in the field.

The well-performed MALDI-TOF analysis  is something that adds really informative data to the literature.

The conclusions are completely consistent with the results.
As highlighted in the already updated comments, the references could be improved with some more recent publications.

Response: We expanded Introduction and added modern references to these sections. In particular, the following references were added:

1) Bryzgunova, O.; Bondar, A.; Ruzankin, P.; Laktionov, P.; Tarasenko, A.; Kurilshikov, A.; Epifanov, R.; Zaripov, M.; Kabilov, M.; Laktionov, P. Locus-Specific Methylation of GSTP1, RNF219, and KIAA1539 Genes with Single Molecule Resolution in Cell-Free DNA from Healthy Donors and Prostate Tumor Patients: Application in Diagnostics. Cancers (Basel). 2021, 13(24), 6234.

2) Camus, V.; Jardin, F. Cell-Free DNA for the Management of Classical Hodgkin Lymphoma. Pharmaceuticals (Basel) . 2021, 14(3), 207.

3) Friedemann, M.; Horn, F.; Gutewort, K.; Tautz, L.; Jandeck, C.; Bechmann, N.; Sukocheva, O.; Wirth, MP.; Fuessel, S.; Menschikowski, M. Increased Sensitivity of Detection of RASSF1A and GSTP1 DNA Fragments in Serum of Prostate Cancer Patients: Optimisation of Diagnostics Using OBBPA-ddPCR. Cancers (Basel). 2021, 13(17), 4459.

4) Hu, G.; Qin, L.; Zhang, X.; Ye, G.; Huang, T. Epigenetic Silencing of the MLH1 Promoter in Relation to the Development of Gastric Cancer and its Use as a Biomarker for Patients with Microsatellite Instability: a Systematic Analysis. Cell Physiol Biochem. 2018, 45(1), 148-162.

5) Hudečková, M.; Koucký, V.; Rottenberg, J.; Gál, B. Gene Mutations in Circulating Tumour DNA as a Diagnostic and Prognostic Marker in Head and Neck Cancer-A Systematic Review. Biomedicines. 2021, 9(11), 1548.

6) Kawamura, Y.; Yamamoto, Y.; Sato, T.-A.; Ochiya, T. Extracellular Vesicles as Trans-Genomic Agents: Emerging Roles in Disease and Evolution. Cancer Sci. 2017, 108(5), 824-830.

7) Kononova, I.V.; Mamaeva, S.N.; Alekseev, V.A.; Nikolaeva, N.A.; Afanasyeva, L.N.; Nikifirov, P.V.; Vasilyeva, N.A.; Vasiliev, I.V.; Maximov, G.V. Simultaneous Detection of the HPV L1 Gene and the Human β-Globin Gene in the Blood Components of Cervical Cancer Patients Living in Yakutia. Int. J. Biomed. 2022, 12(1), 109-114.

8) Li, M.; Xie, S.; Lu, C.; Zhu, L.; Zhu, L. Application of Data Science in Circulating Tumor DNA Detection: A Promising Avenue Towards Liquid Biopsy. Front Oncol. 2021, 11, 692322.

9) Maisano, D.; Mimmi, S.; Dattilo, V.; Marino, F.; Gentile, M.; Vecchio, E.; Fiume, G.; Nisticò, N.; Aloisio, A.; de Santo, M.P.; Desiderio, G.; Musolino, V.; Nucera, S.; Sbrana, F.; Andò, S.; Ferrero, S.; Morandi, A.; Bertoni, F.; Quinto, I.; Iaccino, E. A Novel Phage Display Based Platform for Exosome Diversity Characterization. Nanoscale. 2022, 14(8), 2998-3003.

10) Pu, W.-Y.; Zhang, R.; Xiao, Li.; Wu, Y.-Y.; Gong, W.; Lv, X.-D.; Zhong, F.-Y.; Zhuang, Z.-X.; Bai, X.-M.; Li, K.; Xing, C.-G. Prediction of Cancer Progression in a Group of 73 Gastric Cancer Patients by Circulating Cell-Free DNA. BMC Cancer. 2016, 16(1), 943.

11) Schmidt, B.; Liebenberg, V.; Dietrich, D.; Schlegel, T.; Kneip, C.; Seegebarth, A.; Flemming, N.; Seemann, S.; Distler, J.; Lewin, J.; et al. SHOX2 DNA Methylation is a Biomarker for the Diagnosis of Lung Cancer Based on Bronchial Aspirates. BMC Cancer 2010, 10, 600

12) Warren, J.; Xiong, W.; Bunker, A.; Vaughn, C.; Furtado, L.; Roberts, W.; Fang, J.; Samowitz, W.; Heichman, K. Septin 9 Methylated DNA is a Sensitive and Specific Blood Test for Colorectal Cancer. BMC Med. 2011, 9, 133.

Figures look substantially nice but the related captions should be more informative in order methods and results.

Response: Legends for Figures 1-4 are expanded, figure 1 has been revised.

We trust that we have addressed the reviewers concerns to your satisfaction and we look forward to having our study published in Special Issue "Cell-Free Nucleic Acids: New Insights into Physico-Chemical Properties, Analytical Considerations, and Clinical Applications" of Diagnostics.

Best regards,

Reviewer 2 Report

The manuscript of Tutanov et al. describes characteristics of exosomes isolated from blood plasma of healthy females and breast cancer patients with respect to circulating DNA content and protein composition.

The manuscript is in general nicely written and potentially interesting for experts in the field. However, I have several comments / recommendations / questions:

  1. As the topic is potentially interesting for a general scientific community, the introduction deserves to be or rather must be significantly extended.
  2. What has been published about the cirDNA sequence? Please mention in the manuscript introduction. Would it be in this respect relevant to determine and search for differences in the sequence of cirDNA in HFs and BCPs?
  3. Regarding the statistical significance of the results, I am a bit concerned about the sample size (n=7) and about mixing the exosomes within a group. This should be clearly stated as a limitation of the study after the discussion.
  4. Figure legends must be extended (especially for Figures 2, 3, and 4).
  5. Figures 1a and 1b probably consist of several subfigures. Please clearly mark the borders and improve description so that it is easy to understand for the reader.
  6. Excessive trypsinolysis due to the high Lys contents can be alternatively solved by using a different endopeptidase than trypsin.
  7. Please provide reliability score and number of peptides identified in Table 2.
  8. Line 286: then (not than)
  9. Raw data should be deposited to be available for the reviewers / readers (Mendeley; ProteomeXchange for MS). 

Author Response

Thank you very much for your kind letter, as well as the enclosed suggestions and comments, regarding our manuscript entitled “Blood Plasma Exosomes Contain Circulating DNA in Their Crown”. We thank you for providing critical reviews and helpful suggestions regarding our manuscript. Accordingly, we have carefully revised the paper, highlighted changes within the manuscript, with changes tracked. Please see herein a detailed response to the comments provided, point-by-point, reflecting changes made within the manuscript.

We agree with the opinion of the reviewer and expanded the Introduction by adding information about different sequence of cirDNA in healthy individuals and cancer patients and adding 12 modern relevant references:

1) Bryzgunova, O.; Bondar, A.; Ruzankin, P.; Laktionov, P.; Tarasenko, A.; Kurilshikov, A.; Epifanov, R.; Zaripov, M.; Kabilov, M.; Laktionov, P. Locus-Specific Methylation of GSTP1, RNF219, and KIAA1539 Genes with Single Molecule Resolution in Cell-Free DNA from Healthy Donors and Prostate Tumor Patients: Application in Diagnostics. Cancers (Basel). 2021, 13(24), 6234.

2) Camus, V.; Jardin, F. Cell-Free DNA for the Management of Classical Hodgkin Lymphoma. Pharmaceuticals (Basel) . 2021, 14(3), 207.

3) Friedemann, M.; Horn, F.; Gutewort, K.; Tautz, L.; Jandeck, C.; Bechmann, N.; Sukocheva, O.; Wirth, MP.; Fuessel, S.; Menschikowski, M. Increased Sensitivity of Detection of RASSF1A and GSTP1 DNA Fragments in Serum of Prostate Cancer Patients: Optimisation of Diagnostics Using OBBPA-ddPCR. Cancers (Basel). 2021, 13(17), 4459.

4) Hu, G.; Qin, L.; Zhang, X.; Ye, G.; Huang, T. Epigenetic Silencing of the MLH1 Promoter in Relation to the Development of Gastric Cancer and its Use as a Biomarker for Patients with Microsatellite Instability: a Systematic Analysis. Cell Physiol Biochem. 2018, 45(1), 148-162.

5) Hudečková, M.; Koucký, V.; Rottenberg, J.; Gál, B. Gene Mutations in Circulating Tumour DNA as a Diagnostic and Prognostic Marker in Head and Neck Cancer-A Systematic Review. Biomedicines. 2021, 9(11), 1548.

6) Kawamura, Y.; Yamamoto, Y.; Sato, T.-A.; Ochiya, T. Extracellular Vesicles as Trans-Genomic Agents: Emerging Roles in Disease and Evolution. Cancer Sci. 2017, 108(5), 824-830.

7) Kononova, I.V.; Mamaeva, S.N.; Alekseev, V.A.; Nikolaeva, N.A.; Afanasyeva, L.N.; Nikifirov, P.V.; Vasilyeva, N.A.; Vasiliev, I.V.; Maximov, G.V. Simultaneous Detection of the HPV L1 Gene and the Human β-Globin Gene in the Blood Components of Cervical Cancer Patients Living in Yakutia. Int. J. Biomed. 2022, 12(1), 109-114.

8) Li, M.; Xie, S.; Lu, C.; Zhu, L.; Zhu, L. Application of Data Science in Circulating Tumor DNA Detection: A Promising Avenue Towards Liquid Biopsy. Front Oncol. 2021, 11, 692322.

9) Maisano, D.; Mimmi, S.; Dattilo, V.; Marino, F.; Gentile, M.; Vecchio, E.; Fiume, G.; Nisticò, N.; Aloisio, A.; de Santo, M.P.; Desiderio, G.; Musolino, V.; Nucera, S.; Sbrana, F.; Andò, S.; Ferrero, S.; Morandi, A.; Bertoni, F.; Quinto, I.; Iaccino, E. A Novel Phage Display Based Platform for Exosome Diversity Characterization. Nanoscale. 2022, 14(8), 2998-3003.

10) Pu, W.-Y.; Zhang, R.; Xiao, Li.; Wu, Y.-Y.; Gong, W.; Lv, X.-D.; Zhong, F.-Y.; Zhuang, Z.-X.; Bai, X.-M.; Li, K.; Xing, C.-G. Prediction of Cancer Progression in a Group of 73 Gastric Cancer Patients by Circulating Cell-Free DNA. BMC Cancer. 2016, 16(1), 943.

11) Schmidt, B.; Liebenberg, V.; Dietrich, D.; Schlegel, T.; Kneip, C.; Seegebarth, A.; Flemming, N.; Seemann, S.; Distler, J.; Lewin, J.; et al. SHOX2 DNA Methylation is a Biomarker for the Diagnosis of Lung Cancer Based on Bronchial Aspirates. BMC Cancer 2010, 10, 600

12) Warren, J.; Xiong, W.; Bunker, A.; Vaughn, C.; Furtado, L.; Roberts, W.; Fang, J.; Samowitz, W.; Heichman, K. Septin 9 Methylated DNA is a Sensitive and Specific Blood Test for Colorectal Cancer. BMC Med. 2011, 9, 133.

  1. Regarding the statistical significance of the results, I am a bit concerned about the sample size (n=7) and about mixing the exosomes within a group. This should be clearly stated as a limitation of the study after the discussion.

Response: This article does not focus on tumor markers (in this case, a significant increase in the number of analyzed healthy donors and cancer patients is really required). The aim of this article is to determine the proportion of exoDNA in the plasma cirDNA of healthy women and breast cancer patients, to identify proteins involved in the association of exoDNA with the exosomal membrane, and to propose mechanisms for such an association. Since we could not take 50 ml of blood from a woman to increase the number of molecules detected, for three experiments we used a pooled samples. In particular, individual samples were mixed to generate one pooled sample from HFs and one from BCPs:

  • for cirDNA size estimation by capillary electrophoresis. The information about mixed samples in this experiment was indicated at 7
  • for DNase I treatment. The information about mixed samples in this experiment was indicated at page 3
  • for dot immunoassay. The information about mixed samples in this experiment was indicated at page 3

Individual samples were used:

  • for characterization of exosomes by TEM,
  • for characterization of exosomes by immunogold labeling,
  • for characterization of exosomes by NTA,
  • for estimation of cirDNA concentration
  • for estimation of exoDNA concentration
  • for identification of exosomal protein cargo by MALDI-TOF mass-spectrometry

There are currently very few articles published on proteomics of exosomes from biological samples (not from cell culture). The number of samples analyzed in these articles is comparable to ours or less, due to the high complexity of the analysis (for example, in comparison with the analysis of microRNA using PCR). For example,

Source of exosomes

Number of samples

Refs

saliva

5 healthy donors

Proteomic analysis of microvesicles in human saliva by gel electrophoresis with liquid chromatography-mass spectrometry.

Xiao H, Wong DT.

Anal Chim Acta. 2012;723:61-7.

cerebrospinal fluid

5 healthy donors

Identification and proteomic profiling of exosomes in human cerebrospinal fluid.

Street JM, Barran PE, Mackay CL, et al.

J Transl Med. 2012;10:5.

plasma

14 breast cancer patients

Periostin is identified as a putative metastatic marker in breast cancer-derived exosomes.

Vardaki I, Ceder S, Rutishauser D, et al.

Oncotarget. 2016;7(46):74966-74978.

plasma

6 healthy donors

3 ovarian cancer patients

Proteomics profiling of plasma exosomes in epithelial ovarian cancer: A potential role in the coagulation cascade, diagnosis and prognosis.

Zhang W, Ou X, Wu X.

Int J Oncol. 2019;54(5):1719-1733.

In our study, 7 blood samples from healthy females and 7 blood samples from breast cancer patients were used.

  1. Figure legends must be extended (especially for Figures 2, 3, and 4).

Response: Legends for Figures were expanded.

  1. Figures 1a and 1b probably consist of several subfigures. Please clearly mark the borders and improve description so that it is easy to understand for the reader.

Response: Figure 1 has been improved. Legend for Figure 1 was improved too.

  1. Excessive trypsinolysis due to the high Lys contents can be alternatively solved by using a different endopeptidase than trypsin.

Response: We thank the Reviewer for the technical advice and will take it into account in further work on protein identification.

  1. Please provide reliability score and number of peptides identified in Table 2.

Response: We expanded Table 2 by adding score and number of identified peptides.

  1. Line 286: then (not than)

Response: We thank Reviewer for the identified typo. Typo corrected.

  1. Raw data should be deposited to be available for the reviewers / readers (Mendeley; ProteomeXchange for MS). 

Response: In accordance with the advice of Reviewer, we have deposited raw data from DNA-binding proteins. These data available via link:

https://drive.google.com/drive/folders/1W8ZaFa2di9-lIgTwV9toVV-emEtz9mKS?usp=sharing

Unfortunately, we do not have data for two proteins. One of the researchers died (Dr. T.G. Duzhak) and we do not have an access to her working archive. We can remove these two DNA-binding proteins (CHD5&ZNF835) from Table 2, however, in the future, it may turn out that these proteins are most significant for DNA-binding to the exosome crown. So we decided to leave them. However, if the Reviewer considers it necessary to remove CHD5 and ZNF835 from Table 2, we will do so.

We have no experience in translating this raw data into a format acceptable for uploading to Mendeley or ProteomeXchange. However, we are always glad to send raw data to interested readers upon request.

We trust that we have addressed the reviewers concerns to your satisfaction and we look forward to having our study published in Special Issue "Cell-Free Nucleic Acids: New Insights into Physico-Chemical Properties, Analytical Considerations, and Clinical Applications" of Diagnostics.

Best regards,
